# Nitrates vs. Other Types of Vasodilators and Clinical Outcomes in Patients with Vasospastic Angina: A Propensity Score-Matched Analysis

**DOI:** 10.3390/jcm11123250

**Published:** 2022-06-07

**Authors:** Hyun-Jin Kim, Sang-Ho Jo, Min-Ho Lee, Won-Woo Seo, Hack-Lyoung Kim, Kwan Yong Lee, Tae-Hyun Yang, Sung-Ho Her, Byoung-Kwon Lee, Keun-Ho Park, Youngkeun Ahn, Seung-Woon Rha, Hyeon-Cheol Gwon, Dong-Ju Choi, Sang Hong Baek

**Affiliations:** 1Division of Cardiology, Department of Internal Medicine, Hanyang University College of Medicine, Seoul 04763, Korea; titi8th@hanyang.ac.kr; 2Division of Cardiology, Department of Internal Medicine, Hallym University Sacred Heart Hospital, Anyang-si 14068, Korea; 3Division of Cardiology, Department of Internal Medicine, Soonchunhyang University Seoul Hospital, Seoul 04401, Korea; neoich@gmail.com; 4Division of Cardiology, Department of Internal Medicine, Kangdong Sacred Heart Hospital, Hallym University College of Medicine, Seoul 05355, Korea; wonwooda@gmail.com; 5Cardiovascular Center, Seoul National University Boramae Medical Center, Seoul 07061, Korea; khl2876@gmail.com; 6Division of Cardiology, Seoul St. Mary’s Hospital, The Catholic University of Korea, Seoul 06591, Korea; cycle210@catholic.ac.kr (K.Y.L.); whitesh@catholic.ac.kr (S.H.B.); 7Department of Cardiovascular Medicine, Busan Paik Hospital, Inje University, Busan 04551, Korea; yangthmd@naver.com; 8Department of Cardiovascular Medicine, St. Vincent’s Hospital, The Catholic University of Korea, Suwon 16249, Korea; hhhsungho@naver.com; 9Department of Cardiovascular Medicine, Gangnam Severance Hospital, Yonsei University, Seoul 06273, Korea; cardiobk@gmail.com; 10Division of Cardiology, Department of Internal Medicine, Chosun Medical Center, Gwangju 61453, Korea; keuno21@naver.com; 11Department of Cardiology, Chonnam National University Hospital, Chonnam National University Medical School, Gwangju 61469, Korea; cecilyk@hanmail.net; 12Department of Cardiovascular Medicine, Guro Hospital, Korea University, Seoul 08308, Korea; swrha617@yahoo.co.kr; 13Department of Cardiovascular Medicine, Samsung Medical Center, Sungkyunkwan University, Seoul 06351, Korea; hcgwon@naver.com; 14Division of Cardiology, Department of Internal Medicine, Seoul National University Bundang Hospital, Seongnam 13620, Korea; djchoi@snubh.org

**Keywords:** vasospastic angina, nitrate, vasodilator, acute coronary syndrome

## Abstract

Although vasodilators are widely used in patients with vasospastic angina (VA), few studies have compared the long-term prognostic effects of different types of vasodilators. We investigated the long-term effects of vasodilators on clinical outcomes in VA patients according to the type of vasodilator used. Study data were obtained from a prospective multicenter registry that included patients who had symptoms suggestive of VA. Patients were classified into two groups according to use of nitrates (*n* = 239) or other vasodilators (*n* = 809) at discharge. The composite clinical events rate, including acute coronary syndrome (ACS), cardiac death, new-onset arrhythmia (including ventricular tachycardia and ventricular fibrillation), and atrioventricular block, was significantly higher in the nitrates group (5.3% vs. 2.2%, *p* = 0.026) during one year of follow-up. Specifically, the prevalence of ACS was significantly more frequent in the nitrates group (4.3% vs. 1.5%, *p* = 0.024). After propensity score matching, the adverse effects of nitrates remained. In addition, the use of nitrates at discharge was independently associated with a 2.69-fold increased risk of ACS in VA patients. In conclusion, using nitrates as a vasodilator at discharge can increase the adverse clinical outcomes in VA patients at one year of follow-up. Clinicians need to be aware of the prognostic value and consider prescribing other vasodilators.

## 1. Introduction

Vasospastic angina (VA) is a functional disorder caused by the focal or diffuse spasm of the smooth muscle layer of the coronary arterial wall, resulting in a high grade of obstruction and transient myocardial ischemia [1,2]. Overall, the long-term prognosis of VA is known to be good [3]. However, once a serious heart condition occurs with VA, it may lead to sudden cardiac death following myocardial infarction or fatal ventricular arrhythmia [4,5]. Treatment with calcium channel blockers (CCBs) is recommended as first-line therapy for patients with VA according to current guidelines, since CCBs are highly effective for preventing coronary spasm [1]. Along with CCBs, nitrates or nicorandil are often used as concomitant therapy for the prevention of coronary artery spasm (Class IIa recommendation). Nitrates are metabolized to nitric oxide (NO), which activates NO-cyclic guanosine-3′, -5′-monophasphate (cGMP) signaling pathways within vascular smooth muscle cells, resulting in vasodilation [1,6]. Nicorandil has the properties of nitrates and also acts as a K_ATP_ channel agonist, which could result in vasodilation without intracellular cGMP accumulation [7,8]. Additionally, the vasodilatory effects of other nitrate agents, including molsidomine, involve the main mechanism of NO production and secretion [9]. While these vasodilators can improve vasospastic symptoms acutely, their effects on long-term prognosis in VA patients have been controversial. Some studies suggested that long-term nitrate therapy was neutral to clinical outcomes in patients with VA [10]. Meanwhile, a Japanese multicenter registry [11] demonstrated that long-term nitrate therapy did not improve the clinical prognosis (median follow-up duration 32 months) compared with non-nitrate therapy in patients with VA. Korean data from single-center registry [12] also demonstrated that long-term nitrate therapy worsened prognosis (median follow-up duration 54.7 months). However, these prior research studies have limitations as they included retrospective populations, which may make it unclear whether nitrate promotes poor prognosis or serves as a surrogate marker for more serious heart disease. Indeed, despite the widespread use of nitrates and other vasodilators in patients with VA, there has been no study comparing the effects on long-term prognosis according to vasodilator type. This study investigated the actual prescribing status of vasodilators in VA patients at discharge and effects on prognosis according to the type of vasodilator used in a large-scale nationwide prospective registry.

## 2. Materials and Methods

### 2.1. Study Population

Study data were obtained from a prospective nationwide Vasospastic Angina in Korea registry (VA-Korea). The study design of VA-Korea has been published previously [3,13,14]. Between May 2010 and June 2015, 11 tertiary hospitals in Korea participated in this registry. Patient’s inclusion criteria were: patients were 18 years of age or older, with suspected symptoms of vasospastic angina, and those who underwent invasive coronary angiography (CAG) with ergonovine (EG) provocation test, all of which were satisfied. The exclusion criteria were: end-stage renal disease on continuous dialysis, known malignancy, inflammatory disease, or catheter-induced spasm at baseline. Of 2960 initially enrolled patients with suspected VA (Figure 1), 1987 patients had intermediate or significant spasm after intracoronary EG injection during CAG. Among them, only 1302 patients were prescribed vasodilators when they were discharged: 254 patients were prescribed two or more types of vasodilators as discharge medications and 1048 patients were prescribed one vasodilator. We included the 1048 patients using a single vasodilator in the final analysis and classified the patients into two groups depending on the type of vasodilator used at discharge: nitrates group and other vasodilators group. The other vasodilator group was defined as patients who used nicorandil, molsidomine, or trimetazidine at discharge. This study protocol complied with the Declaration of Helsinki and was reviewed and approved by the Institutional Review Board of Hallym University Sacred Heart Hospital (Approved No. 2010-I007). All patients provided written informed consent prior to study entry.

### 2.2. Data Collection

The patient data were collected through the VA-Korea database via a web-based electronic data capture system containing an electronic case report form. The following patient clinical and demographical characteristics were collected from this database: age, sex, body mass index (BMI; kg/m^2^), blood pressure, cardiovascular risk factors, and previous cardiovascular medications. Laboratory data related with cardiovascular disease were also obtained. We also collected left ventricular ejection fraction from echocardiography data at admission. In addition, we extracted information on the types of vasodilator prescribed at discharge (nitrates, nicorandil, molsidomine, and trimetazidine).

### 2.3. Invasive CAG and EG Provocation Test

The baseline CAG and EG provocation tests were performed according to the Guidelines for Diagnosis and Treatment of Patients with VA of the Japanese Circulation Society [1]. The baseline CAG was performed by a well-trained interventional cardiologist; vasoactive medications were discontinued at least 48 h before the procedure. Intracoronary EG was injected in incremental doses of 20 (E1), 40 (E2), and 60 (E3) μg into the left coronary artery (LCA) for the test of provocation [1,15]. Incremental doses of 10 (E1), 20 (E2), and 40 (E3) μg were injected into the right coronary artery (RCA) when LCA did not induce coronary spasm. When spasm was induced, 200 μg of nitroglycerine was injected. Chest pain, location of spasm and electrocardiography (ECG) change were recorded during the provocation test. ECG change was defined as ST segment depression (≥1 mm) or elevation or T-wave inversion in at least 2 consecutive leads [12]. We defined significant vasospasm as total or luminal diameter narrowing by more than 90% of the coronary arteries accompanied by ECG changes and/or chest pain after EG injection [1]. Intermediate spasm was defined as 50% to 90% luminal diameter stenosis of the coronary arteries. All patients who had spasms on the EG provocation test or spontaneous spasm were treated with medication during follow-up according to the clinician’s discretion.

### 2.4. Study Outcomes

The primary outcome was rate of composite clinical events for one year of follow-up (median duration, 365 days; mean 345.0 ± 60.5 days). The composite clinical events included acute coronary syndrome (ACS), cardiac death, new-onset arrhythmia including ventricular fibrillation (VF) and ventricular tachycardia (VT), and atrioventricular (AV) block. VT was defined as sustained VT resulting in hemodynamic instability, and AV block was defined as a high-degree AV block resulting in hemodynamic instability. All-cause death was also noted during the one-year follow-up. Occurrence of death and the timing of death were confirmed through medical records review or telephone interviews. In addition, readmission or emergency room visits due to angina was investigated for one year after diagnosis with vasospastic angina. To investigate whether patients were taking medication continuously, drug compliance was also assessed for one year after diagnosis. Good compliance was defined as maintaining a vasodilator for one year without any change and interruption, and poor compliance was as discontinuation of a vasodilator within one year.

### 2.5. Statistical Analyses

All categorical data are presented as frequencies and percentages, and continuous variables are expressed as means and standard deviations. For continuous variables, the Shapiro–Wilk test was used for confirming the normal distribution of each dataset. Pearson’s chi-squared test was used to compare categorical variables, the Student’s *t*-test was used to compare normal distributed continuous variables, and the Mann–Whitney U-test was used to compared non-normal distributed continuous variables. In addition, we also used propensity scores and 1:1 matching analysis to adjust the uneven distribution of baseline characteristics between the nitrate group and other vasodilator groups. A multiple logistic regression model was constructed to represent the propensity score, which was the probability of the nitrates group. The adjusted variables were as follows: age, sex, history of coronary artery disease, hypertension, and diabetes, current smoking status, alcohol drinking, and cardiovascular medications. The 174 patients in the nitrates group were matched to 174 patients in the other vasodilators group. McNemar’s test was used to compare categorical variables between matched patient groups, and a paired *t*-test was used for continuous variables. Kaplan–Meier survival analysis and log-rank test were used to compare the ACS-free survival rates and cumulative composite clinical events-free survival rates between the nitrates group and the other vasodilators group. In addition, univariate analysis and multivariate logistic regression analysis were performed to evaluate the risk of ACS after adjustment for individual risk factors. Variables with predictive significance (*p* < 0.05) of ACS in univariate analysis were included in the regression analysis. A *p*-value less than 0.05 was considered statistically significant. All analyses were performed using SPSS 21.0 software (IBM Corp., Armonk, NY, USA).

## 3. Results

### 3.1. Baseline Characteristics

Among 1048 patients with VA using a single vasodilator at discharge who underwent CAG and EG provocation test, there were 239 patients who were prescribed a nitrate at discharge (nitrates group) and 809 patients who were prescribed another vasodilator (other vasodilators group: 521 patients used nicorandil, 177 patients used molsidomine, and 111 patients used trimetazidine). Patients’ baseline characteristics according to vasodilator type are shown in Table 1. Patients in the nitrates group were significantly older than patients in the other vasodilators group, and significantly more patients in the nitrates group reported alcohol drinking and current smoking than patients in the other vasodilators group. Previous use of antiplatelet agents and statins was more frequent in the other vasodilators group than in the nitrates group. Table 1 also shows the laboratory findings of the two groups: there were no significant differences between the two groups. Likewise, there were no significant differences in other histories or medications related to traditional cardiovascular risk factors or diseases between the two groups. Appendix A online) shows the comparison of coronary angiographic characteristics after EG provocation test between the two groups. There were no significant differences in location of spasm between the two groups, but provocation-associated chest pain was more frequent in the nitrates group than in the other vasodilators group. In addition, there was no significant difference in multi-vessel involvement in which spam occurred in two or more coronary arteries after EG provocation test: 23.1% in the nitrates group and 26.7% in the other type of vasodilator group (*p* = 0.308).

### 3.2. Clinical Outcomes according to Vasodilator Type

Among 1048 patients, 780 patients had one year of follow-up data, and the composite clinical events of ACS, cardiac death, VT or VF, or AV block occurred in 23 patients. The one-year composite clinical events rate was significantly higher in the nitrates group than in the other vasodilators group (5.3% vs. 2.2%, *p* = 0.026) (Table 2). Specifically, the prevalence of one-year ACS was significantly more frequent in the nitrates group than in the other vasodilators group (4.3% vs. 1.5%, *p* = 0.024). However, one-year all-cause death rates did not differ significantly according to the vasodilator type. There was also no significant difference between the two groups in terms of readmission or emergency room visits for one year. Based on whether the VA patients received a nitrate or other vasodilators at discharge, the cumulative composite clinical events rate and the cumulative ACS-free survival rate were analyzed, and results are shown in Figure 2A,B. Patients in the nitrates group had a significantly lower cumulative event-free survival rate than patients in the other vasodilators group at the one-year follow-up (89.2% vs. 96.1%, log-rank *p* = 0.026) (Figure 2A). Patients in the nitrates group also had a significantly lower cumulative ACS-free survival rate (90.4% vs. 97.1%, log-rank *p* = 0.023) (Figure 2B) (Appendix A showed the time to event of each individuals). Additionally, there was no significant difference in the rate of the one-year composite clinical events among nicorandil group, molsidomine group, and trimetazidine group (Appendix A). The prevalence of one-year ACS showed also no significant difference among three groups.

### 3.3. Clinical Outcomes in Propensity Score-Matched Population

After propensity score matching, 174 patients in the nitrates group were successfully matched to an equal number of patients in the other vasodilators group. Baseline characteristics were not significantly different between groups after propensity score matching (Appendix A). The rate of one-year composite clinical events of the matched population was significantly higher in the nitrates group (5.7% vs. 1.1%, *p* = 0.035) (Table 3). In addition, the one-year ACS events rate of the matched population was significantly higher in the nitrates group (4.6% vs. 0.6%, *p* = 0.037). Figure 3 shows the cumulative composite clinical events rate and cumulative ACS-free survival rate between the matched groups. Patients in the nitrates group had a significantly lower cumulative event-free survival rate than patients in the other vasodilators group (94.2% vs. 98.9%, log-rank *p* = 0.021) (Figure 3A), as well as a lower cumulative ACS-free survival rate (95.4% vs. 99.4%, log-rank *p* = 0.019) (Figure 3B).

### 3.4. Effect of Nitrate Type on One-Year ACS Rate in VA Patients

According to univariate analysis (Table 4), the following factors were associated with ACS events at one-year follow-up in VA patients: use of nitrates at discharge (odds ratio (OR), 2.86; 95% confidence interval (CI), 1.104–7.420; *p* = 0.031) and age. After adjusting for age, the Cox regression analysis showed that the use of nitrates at discharge was independently associated with a 2.69-fold increased hazard for ACS in VA patients (OR, 2.69; 95% CI, 1.035–6.979; *p* = 0.042). However, the use of other vasodilators, including nicorandil, molsidomine, and trimetazidine, at discharge was not an independent predictor of ACS in VA patients.

### 3.5. Subgroup Analysis

A subgroup analysis of the one-year clinical events rate of patients with VA according to drug compliance was performed. There were 776 patients with confirmed one-year drug compliance: 55.9% of patients in the nitrates group maintained the nitrate for one year, and 65.6% of patients in the other vasodilators group maintained the vasodilator for one year. Among patients with good drug compliance during one year, there were no significant differences in composite clinical events rate or all-cause death rate between the two groups (Appendix A). However, among patients with poor compliance, the one-year ACS rate was significantly higher in the nitrates group than in the other vasodilators group (7.3% vs. 2.0%, *p* = 0.036) (Appendix A).

## 4. Discussion

According to results from this nationwide prospective large-scale registry, the incidence of one-year composite clinical events including ACS was significantly higher in VA patients who used nitrates at discharge than in those who used other vasodilators at discharge; the adverse effects of nitrates were consistent after propensity score matching. Specifically, the use of nitrates at discharge was independently associated with a 2.69-fold increased risk of ACS in patients with VA. The nitrates group had lower drug compliance during one year of follow-up compared to the other vasodilators group, which affected one-year clinical events rates. Indeed, in patients with poor compliance, the one-year ACS rate in the group who used nitrates at discharge was significantly higher than in the group who used other vasodilators at discharge.

Nitrates, nicorandil, and other types of vasodilators are widely used for relieving acute angina symptoms in ischemic heart disease, including VA [16]. Long-acting nitrates are metabolized to NO within vascular smooth muscle cells, resulting in dilation of the coronary vasculature [1,6]. Nitrates including isosorbide dinitrate or isosorbide mononitrate ER have been proven to suppress acute angina symptoms and prevent recurrent attacks [17]. However, the frequent and continued use of nitrates can cause reduced vasodilatory effects due to the development of nitrate tolerance, which is caused by multiple factors [18,19]. In addition, during periods of nitrate withdrawal or nitrate-free periods, “rebound angina” may occur, in which the frequency of angina increases suddenly [16,20]. Increased sensitivity to vasoconstriction has been known to be a possible mechanism to explain rebound angina, while the vasodilating effect of NO decreases during the nitrate-free periods [20,21]. This is consistent with our results that the nitrates group had an increased risk of ACS during one year of follow-up, especially VA patients with poor drug compliance. On the contrary, nicorandil, which has properties similar to those of nitrates and acts as a K_ATP_ channel agonist, does not cause tolerance or rebound angina [16]. This is because nicorandil opens up potassium channels in the plasma membrane with the hyperpolarization of plasma smooth muscle cells, which can cause vascular relaxation without cGMP accumulation in the cells [22]. The role of K_ATP_ channels is to inhibit the formation of cGMP, which is associated with nitrate tolerance [22]. Another vasodilator, molsidomine, is a NO donor and delivers NO directly to vascular smooth muscle cells, activating the soluble guanylate cyclase, which synthesizes vasodilating cGMP from guanosine triphosphate [23,24]. This may also be the reason for the lower levels of tolerance to molsidomine compared to other nitrates. Since the mechanisms involved in vasodilation differ according to the type of drug, rebound angina or drug tolerance can occur differently in patients with VA. In this study, the composite clinical events rate including ACS was significantly higher in the nitrates group than in the other vasodilators group during one year of follow-up, which was maintained even after propensity score matching. Moreover, other vasodilators, including nicorandil, molsido-mine, and trimetazidine, did not raise ACS risk. Although the exact mechanism for different clinical outcomes according to type of vasodilator in VA patients is unclear, it may be related to the different endothelium-dependent responsiveness of vascular smooth muscle cell, which is an important pathogenesis of VA. To the best of our knowledge, there have been no clinical or experimental studies of direct comparison of this issue; a large-scale study will be needed in the future.

There was a small study that evaluated the long-term effects of nitrate treatment on cardiac events including cardiac death and readmission for ACS in VA patients who were treated with CCBs in a single Japanese center [25]. There were 48 patients who were treated with nitrates, 38 who were treated with nicorandil, and 145 patients who did not use vasodilators. The results showed that nitrates independently increased the risk of cardiac events by 5.18 times during 70 months of follow-up, but nicorandil did not increase the risk. In a recent multicenter study in Japan, Takahashi et al. [11] showed a long-term effect of nitrate therapy on major adverse cardiac events (MACE), including non-fatal myocardial infarction, cardiac death, heart failure, hospitalization due to unstable angina, and appropriate implantable cardioverter defibrillator shocks in 1492 VA patients. When they were followed for a median 32 months, the nitrates group did not have a significantly decreased or increased risk of MACE compared with the group not using nitrates. Even when nitrates and nicorandil were analyzed separately, neither of them affected MACE. In another recent study in Korea, Kim et al. [26] revealed that nitrates increased the risk of MACE, including cardiac death, myocardial infarction, any revascularization, or readmission due to recurrent angina, by 1.32-fold compared with not using nitrates in patients with VA during a median 55 months of follow-up. Specifically, patients treated with nitrates had a significantly higher risk of MACE by 1.70-fold, but nicorandil did not show any association with an increased risk of MACE. Although the clinical outcomes of each previous study were different from the outcomes of our study, which was composite clinical events including ACS, cardiac death, new-onset arrhythmia including VT and VF, and AV block, they showed that nicorandil had a neutral effect on adverse clinical outcomes and nitrates had a neutral effect or a tendency to increase the risk of adverse clinical outcomes. The most recent study using this VA-Korea registry [27] also showed that the risk of ACS at 2 years was significantly increased in the nitrate group compared with the non-vasodilator group (HR 2.49, 95% CI 1.01–6.14, *p* = 0.047) and that was not increased in the non-nitrate other type vasodilator group compared with the non-vasodilator group (HR 0.92, 95% CI 0.39–2.13, *p* = 0.841). However, composite clinical outcome including ACS, cardiac death, and new-onset arrhythmia at 2 years showed no significant differences between the nitrate, non-nitrate other type vasodilator, or non-vasodilator groups. This is not a direct comparison study of nitrate and non-nitrate other types of vasodilators. In this regard, it is notable that our study directly compared nitrates to other vasodilators and also presented and analyzed drug compliance.

The results of this study can be helpful in real-world practice. When VA patients are prescribed nitrates at discharge, their drug compliance may be poor owing to the side effects such as headache or dizziness. In the nitrates group, 55.9% maintained the drug for one year, and in the other vasodilators group, 65.6% maintained the drug. Poor drug compliance with nitrates will increase the occurrence of rebound angina and may also be associated with an increased risk of adverse clinical events compared to other vasodilators. Clinicians should be able to select a differentiated drug for each individual considering the effect of drug compliance to vasodilators on the prognosis in VA patients. In addition, vasospasm was confirmed by using EG in this study, but acetylcholine also can induce spasm, and that effect is promptly dissolved by intracoronary nitroglycerin. Since this study is about the vasodilator effect on clinical outcome in vasospastic patients, further studies are needed, but it is carefully suggested that the results can be applied to vasospastic angina patients who have been diagnosed with acetylcholine.

Several limitations of this study should be considered. First, this is a prospective multicenter cohort study, and it may have inevitable bias that can affect the results unlike a randomized controlled trial. However, to avoid bias as much as possible, propensity score matching and multivariate logistic regression analysis were attempted. Second, regardless of the type of vasodilator prescribed, just over half of the VA patients maintained the use of a vasodilator for one year; this may have affected our results because patients may not have had sufficient effectiveness of the nitrates or other vasodilators. Third, the patients in this study included both intermediate spasm or significant spasm after EG provocation test, and it was not a study targeting only definite vasospastic angina, but all patients with vasospasm of 50% or more who needed vasodilator therapy under the judgment of the clinician. However, even if the EG provocation test shows intermediate vasospasm results, it does not mean that vasodilator is not used in real clinical practice. This study showed the prescription patterns of vasodilator and its effects on prognosis according to the type of vasodilator in all vasospastic angina patients who need vasodilator in real clinical settings. Finally, subgroup analysis results were obtained from a much smaller sample size that was analyzed according to drug compliance, and this may limit the interpretation of this results.

## 5. Conclusions

In conclusion, prescribing nitrates as a vasodilator at discharge in VA patients can increase the adverse clinical outcomes including ACS during one year; poor compliance with nitrates is also associated with adverse clinical outcomes. This is an emphatic real clinical practice and prognosis for patients with symptoms suspected of VA who underwent CAG and EG tests. Thus, in the management of VA patients, clinicians should choose a vasodilator in consideration of a patient’s compliance, as well as the drug mechanism, and they should consider prescribing vasodilators other than nitrates in order to achieve better clinical outcomes. In addition to this study, landmark trials that can provide a guide for prescribing vasodilators in VA patients will be needed in the future, and if these evidences are accumulated, it will form the basis for the management of VA patients.

## Figures and Tables

**Figure 1 jcm-11-03250-f001:**
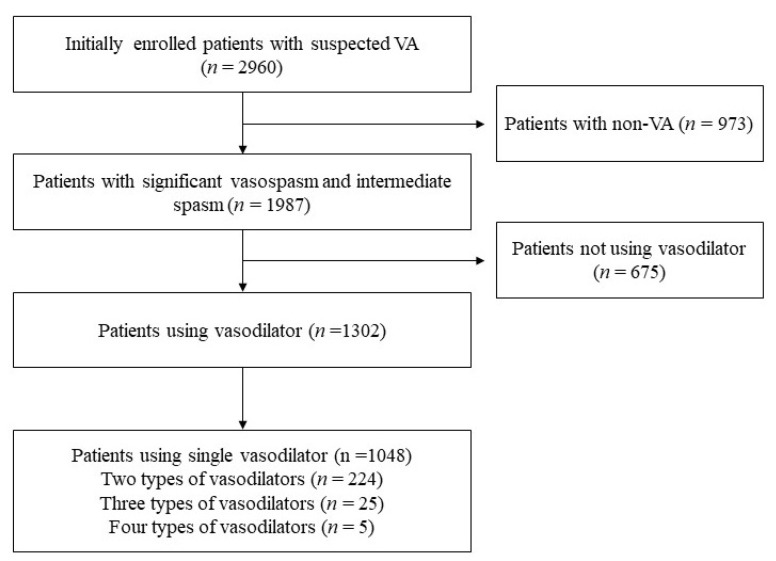
Study population selection process.

**Figure 2 jcm-11-03250-f002:**
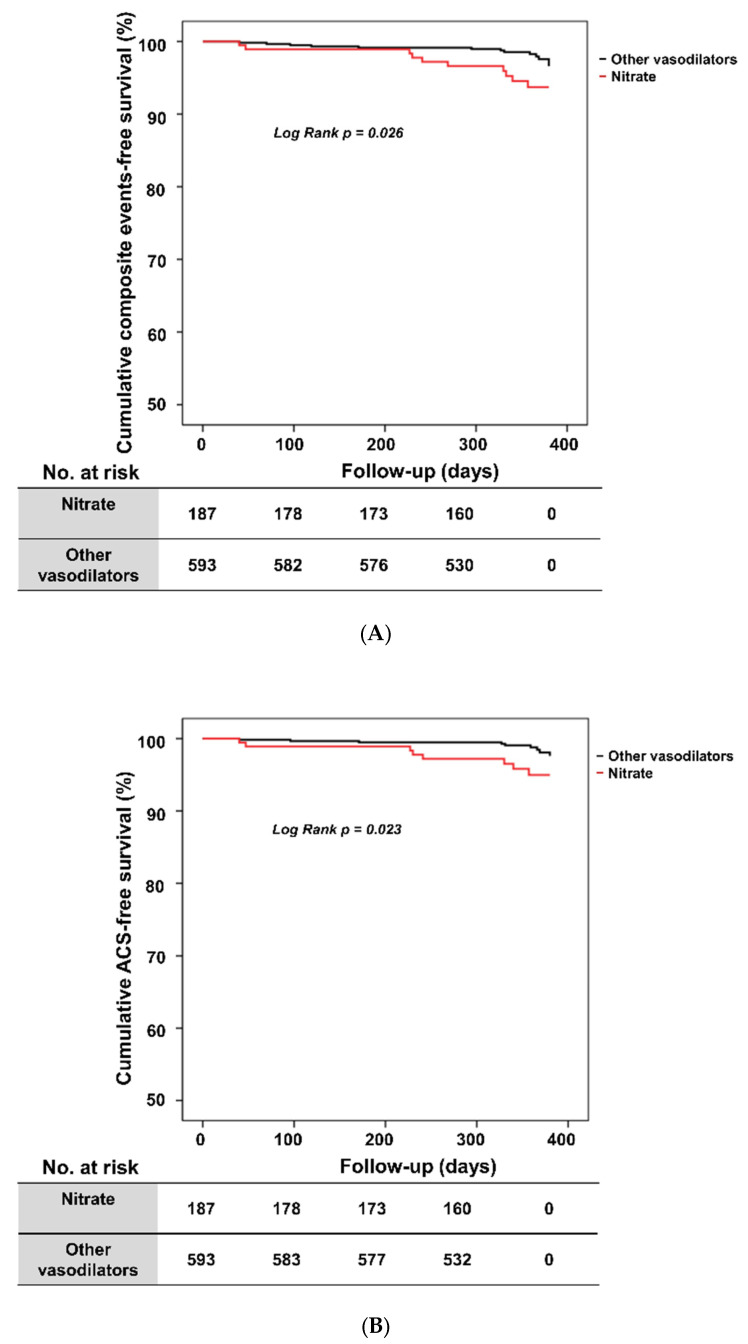
Kaplan–Meier survival curves of the entire population. (**A**) Cumulative composite events-free survival according to vasodilator. (**B**) Cumulative ACS-free survival according to vasodilator. ACS, acute coronary syndrome.

**Figure 3 jcm-11-03250-f003:**
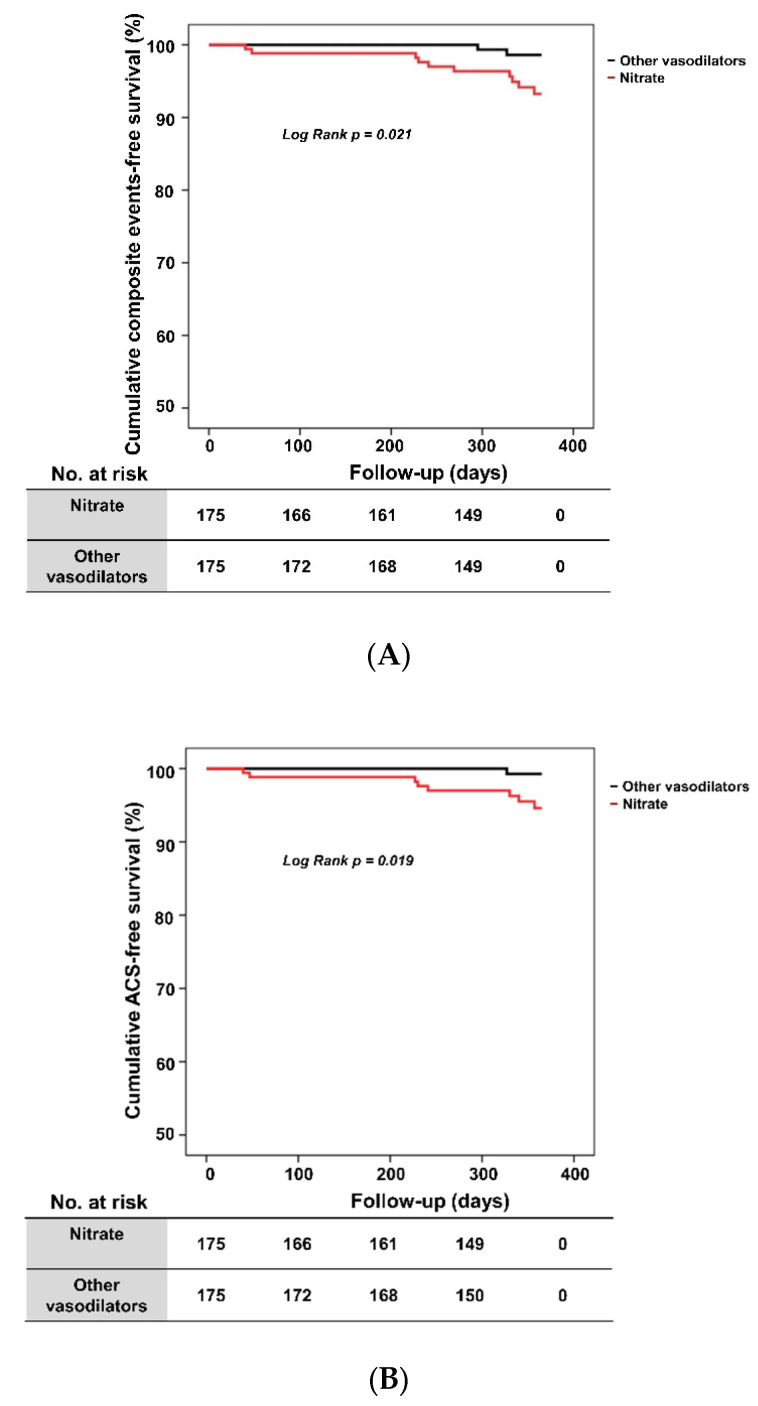
Kaplan–Meier survival curves in propensity score-matched population. (**A**) Cumulative composite events-free survival according to vasodilator. (**B**) Cumulative ACS-free survival according to vasodilator. ACS, acute coronary syndrome.

**Table 1 jcm-11-03250-t001:** Baseline Characteristics.

	All (*n* = 1048)	Nitrates (*n* = 239)	Other Types of Vasodilator (*n* = 809)	*p* Value
Age, years	54.8 ± 11.2	52.6 ± 11.4	55.5 ± 11.1	0.001
Male, *n* (%)	666 (63.5)	160 (66.9)	5056 (62.5)	0.214
BMI, kg/m^2^	24.7 ± 3.3	24.9 ± 4.1	24.7 ± 3.1	0.450
SBP, mmHg	126.0 ± 18.0	126.8 ± 18.7	125.7 ± 17.8	0.404
DBP, mmHg	77.2 ± 12.2	78.3 ± 13.3	76.9 ± 11.8	0.118
Previous CAD, *n* (%)	108 (10.3)	19 (7.9)	89 (11.0)	0.171
Diabetes mellitus, *n* (%)	101 (9.6)	20 (8.4)	81 (10.0)	0.446
Hypertension, *n* (%)	386 (36.9)	98 (41.0)	288 (35.6)	0.131
Dyslipidemia, *n* (%)	183 (17.5)	46 (19.4)	137 (17.0)	0.382
Alcohol drinking, *n* (%)	455 (43.4)	137 (57.3)	318 (39.3)	<0.001
Current smoking, *n* (%)	304 (29.5)	91 (38.1)	213 (26.9)	0.001
**Laboratory finding**				
Hemoglobin, g/dL	13.9 ± 1.9	13.9 ± 1.8	13.9 ± 1.9	0.945
Creatinine, mg/dL	0.8 ± 0.2	0.8 ± 0.2	0.8 ± 0.2	0.294
Glucose, mg/dL	111.3 ± 35.8	112.8 ± 46.4	110.9 ± 32.1	0.565
hs-CRP, mg/dL	0.9 ± 5.8	1.1 ± 7.0	0.8 ± 5.3	0.453
Total cholesterol, mg/dL	173.6 ± 35.6	175.3 ± 35.2	173.0 ± 35.7	0.406
LDL cholesterol, mg/dL	103.1 ± 31.8	103.6 ± 31.4	103.0 ± 31.9	0.811
Triglyceride, mg/dL	145.7 ± 105.4	151.5 ± 94.7	143.9 ± 108.6	0.349
HDL cholesterol, mg/dL	46.3 ±12.7	42.2 ± 11.9	46.7 ± 13.0	0.126
LV EF, %	64.6 ± 6.6	65.1 ± 6.1	64.4 ± 6.8	0.167
**Previous cardiovascular medication**				
Antiplatelet, *n* (%)	222 (21.3)	37 (15.5)	186 (23.0)	0.042
Statin, *n* (%)	163 (15.6)	26 (10.9)	137 (16.9)	0.025
CCB, *n* (%)	191 (18.2)	40 (16.7)	151 (18.7)	0.166
**Discharge medication**				
CCB, *n* (%)	959 (91.5)	220 (92.1)	739 (91.3)	0.732
**Clinical diagnosis before ergonovine**				
Angina, *n* (%)	962 (92.1)	226 (94.6)	736 (91.4)	0.114
Myocardial infarction, *n* (%)	18 (1.7)	3 (1.3)	15 (1.9)	0.777
Cardiac arrest, *n* (%)	11 (1.1)	6 (2.5)	5 (0.6)	0.022
Syncope, *n* (%)	11 (1.1)	4 (1.7)	7 (0.9)	0.286
VT or VF, *n* (%)	5 (0.5)	1 (0.4)	4 (0.5)	1.000
AV block, *n* (%)	1 (0.1)	0 (0.0)	1 (0.1)	1.000

AV, atrioventricular; BMI, body mass index; CAD, coronary artery disease; CCB, calcium channel blocker; DBP, diastolic blood pressure; HDL, high-density lipoprotein; hs-CRP, high sensitive-C reactive protein; LDL, low-density lipoprotein; LV EF, left ventricular ejection fraction; SBP, systolic blood pressure; VF, ventricular fibrillation; VT, ventricular tachycardia.

**Table 2 jcm-11-03250-t002:** One-year clinical event rate of patients with VA according to types of vasodilators.

	All (*n* = 780)	Nitrates (*n* = 187)	Other Types of Vasodilator (*n* = 593)	*p* Value
Composite events	23 (2.9)	10 (5.3)	13 (2.2)	0.026
ACS	17 (2.2)	8 (4.3)	9 (1.5)	0.024
Cardiac death	1 (0.1)	0 (0.0)	1 (0.1)	0.567
VT or VF	2 (0.3)	1 (0.5)	1 (0.2)	0.422
AV block	3 (0.4)	1 (0.5)	2 (0.3)	0.561
All-cause death	3 (0.4)	1 (0.5)	2 (0.3)	0.561
Readmission or emergency room visits due to angina	88 (11.3)	23 (12.3)	65 (11.0)	0.614

ACS, acute coronary syndrome; AV, atrioventricular; VA, vasospastic angina; VF, ventricular fibrillation; VT, ventricular tachycardia.

**Table 3 jcm-11-03250-t003:** One-year clinical event rate of patients with VA according to types of vasodilators after 1:1 propensity-matching.

	All (*n* = 348)	Nitrates (*n* = 174)	Other Types of Vasodilator (*n* = 174)	*p* Value
Composite events	12 (3.4)	10 (5.7)	2 (1.1)	0.035
ACS	9 (2.6)	8 (4.6)	1 (0.6)	0.037
Cardiac death	0 (0.0)	0 (0.0)	0 (0.0)	-
VT or VF	1 (0.3)	1 (0.6)	0 (0.0)	1.000
AV block	2 (0.6)	1 (0.6)	1 (0.6)	1.000
All-cause death	2 (0.6)	1 (0.6)	1 (0.6)	1.000
Readmission or emergency room visits due to angina	43 (12.4)	22 (12.6)	21 (12.1)	0.871

ACS, acute coronary syndrome; AV, atrioventricular; VA, vasospastic angina; VF, ventricular fibrillation; VT, ventricular tachycardia.

**Table 4 jcm-11-03250-t004:** Predictors of ACS in patients with VA.

	Univariate			Multivariate		
	OR	95% CI	*p*	OR	95% CI	*p*
Nitrate	2.86	1.104–7.420	0.031	2.69	1.035–6.979	0.042
Nicorandil	1.10	0.424–2.847	0.847	-	-	-
Molsidomine	0.04	0.000–11.521	0.263	-	-	-
Trimetazidine	0.04	0.000–41.670	0.368	-	-	-
Age	0.96	0.914–1.000	0.049	0.96	0.915–1.003	0.067
Previous CAD	1.06	0.242–4.641	0.938	-	-	-
Hypertension	0.85	0.315–2.307	0.754	-	-	-
Diabetes	1.43	0.328–6.276	0.632	-	-	-
Current smoking	1.51	0.665–3.406	0.327	-	-	-
Alcohol drinking	1.01	0.452–2.242	0.987	-	-	-
LDL-cholesterol	1.01	0.995–1.026	0.193	-	-	-
CCB at index admission	1.49	0.198–11.268	0.697	-	-	-

ACS = acute coronary syndrome; CAD = coronary artery disease; CCB = calcium-channel blocker; CI = confidence interval; LDL = low-density lipoprotein; OR = odds ratio; VA = vasospastic angina.

## Data Availability

Not applicable.

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
