# Peer review of "Nitrates vs. Other Types of Vasodilators and Clinical Outcomes in Patients with Vasospastic Angina: A Propensity Score-Matched Analysis"

_jcm, 2022, doi:10.3390/jcm11123250_

Round 1
Reviewer 1 Report
The article is very complete and well-written, with deep information on the subject, as Authors previously published on the same topic. I believe it will be of great importance for researchers/ clinicians in the field. However, the reference to previous works published by this team (including the results of a 2-year follow-up J Am Heart Assoc. 2022 Apr 5;11(7):e023776. doi: 10.1161/JAHA.121.023776 and others) have been somehow omitted. Please consider the latter and other relevant work when discussing the results and discussions.
Author Response
Point 1: The article is very complete and well-written, with deep information on the subject, as Authors previously published on the same topic. I believe it will be of great importance for researchers/ clinicians in the field. However, the reference to previous works published by this team (including the results of a 2-year follow-up J Am Heart Assoc. 2022 Apr 5;11(7):e023776. doi: 10.1161/JAHA.121.023776 and others) have been somehow omitted. Please consider the latter and other relevant work when discussing the results and discussions.
[Response]
Thank you for your important comment. We fully agree with reviewer’s comment and we added the most current reference “27. Lim, Y.; Kim, M.C.; Ahn, Y.; Cho, K.H.; Sim, D.S.; Hong, Y.J.; Kim, J.H.; Jeong, M.H.; Baek, S.H.; Her, S.H.; et al. Prognostic Impact of Chronic Vasodilator Therapy in Patients With Vasospastic Angina. J Am Heart Assoc 2022, 11, e023776, doi:10.1161/JAHA.121.023776”.
This article was published at April 2022, so we did not added this reference before we first submitted the original version of manuscript. According to this reference in revised version of manuscript, we added the sentence about this article and discussed about it as below. (page 13, line 349-356).
“Most recent study using this VA-Korea registry [27] also showed that the risk of ACS at 2 years was significantly increased in the nitrate group compared with the non-vasodilator group [HR 2.49, 95% CI 1.01-6.14, p = 0.047] and that was not increased in the non-nitrate other type vasodilator group compared with the non-vasodilator group [HR 0.92, 95% CI 0.39-2.13, p = 0.841]. However, composite clinical outcome including ACS, cardiac death, and new-onset arrhythmia at 2 years showed no significant differences between the nitrate, non-nitrate other type vasodilator, or non-vasodilator groups. This is not a direct comparison study of nitrate and non-nitrate other types of vasodilators.”
Thank you again your comment.

Reviewer 2 Report
The “Nitrates vs. Other Types of Vasodilators and Clinical Outcomes
in Patients with Vasospastic Angina” was investigated by Kim et al. Although the work is scientifically sound, several major flows must be addressed.
Abstract
1. The purpose and importance of this research must be expressed more effectively in the abstract.
Introduction
1. Some of the references in the introduction section are out of date. The introduction section appears to be lacking important information. This section needs to be completely rewritten to include more recent references.
2. Try to incorporate prior research limitations as well, and how the current study addresses such constraints.
3. The research gaps should be explicitly specified, along with any important references.
4. The rationale for only one year of patient follow-up should be explained.
5. Avoid to use the words such as We/Our.
Materials and Methods
1. Include information on the patient’s inclusion and exclusion criteria in the study population.
2. Include information regarding the patient consent form.
3. Make figure 1's title better.
4. In the materials and methods section, assign sub-section numbers to all of the titles.
Results and Discussion
1. The result part is well written. However, the references in the discussion are out of date and inappropriate.
2. Instead of describing more of a literature background, I would advise that the authors refocus their discussion to illustrate how the outcomes of their work fit into the greater picture of what is current today.
3. In the discussion, avoid using the words Our/We (eg. Our Study, We tried and We Showed etc.).
Conclusions
1. The study's limitations must be mentioned.
2. Future perspectives must be addressed in the end.
3. The importance of the findings in this study should be emphasized by the author.
References
1. Most of the references are out of date. In the introduction and discussion sections, the authors must refer and include the most recent references.
Round 2
Reviewer 2 Report
The authors revised the content of the manuscript and addressed all of the comments. As a result, I recommend that it be published in JCM in its present form.